# The lncRNA landscape of breast cancer reveals a role for DSCAM-AS1 in breast cancer progression

Yashar S. Niknafs[1,2,*], Sumin Han[3,*], Teng Ma[3,4], Corey Speers[3,5,6], Chao Zhang[3], Kari Wilder-Romans[3], Matthew K. Iyer[1,7], Sethuramasundaram Pitchiaya[1], Rohit Malik[1], Yasuyuki Hosono[1], John R. Prensner[1], Anton Poliakov[1], Udit Singhal[1,8], Lanbo Xiao[1], Steven Kregel[1], Ronald F. Siebenaler[1], Shuang G. Zhao[3], Michael Uhl[9], Alexander Gawronski[10], Daniel F. Hayes[5,6,11], Lori J. Pierce[3,5,6], Xuhong Cao[1,8], Colin Collins[13], Rolf Backofen[9], Cenk S. Sahinalp[10,12,13], James M. Rae[5,6,11], Arul M. Chinnaiyan[1,2,5,6,8,14,15,**] & Felix Y. Feng[1,3,5,6,**,†]

Molecular classification of cancers into subtypes has resulted in an advance in our understanding of tumour biology and treatment response across multiple tumour types. However, to date, cancer profiling has largely focused on protein-coding genes, which comprise <1% of the genome. Here we leverage a compendium of 58,648 long noncoding RNAs (lncRNAs) to subtype 947 breast cancer samples. We show that lncRNA-based profiling categorizes breast tumours by their known molecular subtypes in breast cancer. We identify a cohort of breast cancer-associated and oestrogen-regulated lncRNAs, and investigate the role of the top prioritized oestrogen receptor (ER)-regulated lncRNA, *DSCAM-AS1*. We demonstrate that *DSCAM-AS1* mediates tumour progression and tamoxifen resistance and identify hnRNPL as an interacting protein involved in the mechanism of *DSCAM-AS1* action. By highlighting the role of *DSCAM-AS1* in breast cancer biology and treatment resistance, this study provides insight into the potential clinical implications of lncRNAs in breast cancer.

[1] Michigan Center for Translational Pathology, University of Michigan, Ann Arbor, Michigan 48109, USA. [2] Department of Cellular and Molecular Biology, University of Michigan, Ann Arbor, Michigan 48109, USA. [3] Department of Radiation Oncology, University of Michigan, Ann Arbor, Michigan 48109, USA. [4] Department of Radiation Toxicology and Oncology, Beijing Key Laboratory for Radiobiology (BKLRB), Beijing Institute of Radiation Medicine, Beijing 100850, P. R. China. [5] Breast Oncology Program, University of Michigan, Ann Arbor, Michigan 48109, USA. [6] Comprehensive Cancer Center, University of Michigan, Ann Arbor, Michigan 48109, USA. [7] Department of Computational Medicine and Bioinformatics, Ann Arbor, Michigan 48109, USA. [8] Howard Hughes Medical Institute, University of Michigan, Ann Arbor, Michigan 48109, USA. [9] Department of Computer Science and Centre for Biological Signaling Studies (BIOSS), University of Freiburg, Freiburg 79110, Germany. [10] School of Computing Science, Simon Fraser University, Burnaby, British Columbia, Canada V5A 1S6. [11] Department of Internal Medicine, University of Michigan, Ann Arbor, Michigan 48109, USA. [12] School of Informatics and Computing, Indiana University, Bloomington, Indiana 47405, USA. [13] Vancouver Prostate Centre, Vancouver, British Columbia, Canada V6H 3Z6. [14] Department of Pathology, University of Michigan, Ann Arbor, Michigan 48109, USA. [15] Department of Urology, University of Michigan, Ann Arbor, Michigan 48109, USA. * These authors contributed equally to this work. ** These authors jointly supervised this work. † Present address: Departments of Radiation Oncology, Urology, and Medicine, Helen Diller Family Comprehensive Cancer Center, University of California at San Francisco, San Francisco, California 94115, USA. Correspondence and requests for materials should be addressed to A.M.C. (email: arul@umich.edu) or to F.Y.F. (email: felix.feng@ucsf.edu).

Long noncoding RNAs (LncRNAs) have recently been implicated in a variety of biological processes, including carcinogenesis and tumour growth[1–6]. Operating through a myriad of mechanisms[2], lncRNAs have challenged the central dogma of molecular biology as prominent functional RNA molecules. To investigate the role of lncRNAs in breast cancer, we interrogated the expression of lncRNAs across an RNA-sequencing (RNA-seq) breast tissue cohort comprised of 947 breast samples[7,8]. Previously, in a large-scale *ab initio* meta-assembly study from 6,503 RNA-seq libraries, we discovered ~45,000 of unannotated human lncRNAs[7], and this assembly was utilized for the present study. Building on prior work that has begun to investigate the role of lncRNAs in breast cancer[9], we set out to perform a comprehensive analysis of breast cancer tissue RNA-seq data to identify the lncRNAs potentially involved in breast cancer.

Patients with oestrogen receptor (ER)-positive breast cancer have better prognosis than those with ER-negative disease, based on both a more indolent natural history but perhaps more importantly due to effective anti-oestrogen, also designated 'endocrine,' therapy[10]. Despite the efficacy of endocrine therapy, however, the majority of breast cancer deaths occur in women with ER-positive breast cancers, because the incidence of ER-positive versus-negative disease is much higher (approximately 80 versus 20%), and because a substantial fraction of women either have inherent or acquired endocrine therapy-resistant disease[11].

Taken together, these considerations highlight the pressing need to understand the biology of the ER-driven breast cancers and their mechanism of resistance to endocrine therapy. The mechanism through which ER mediates cancer initiation and progression is an area of intense scientific investigation[12–14] that remains incompletely understood. In this regard, while substantial research has been focused on ER abnormalities, such as mutations in the gene encoding for ER (*ESR1*)[14,15] and on the co-existing activation pathways that might mediate resistance, such as HER2[16], few studies exist that interrogate ER-regulated noncoding RNAs[17–21]. Therefore, we set out to perform a comprehensive discovery and investigation of those lncRNAs that are driven by oestrogen in breast cancers drawing from a large human tissue RNA-seq cohort.

## Results

### Identification of ER- and breast cancer-associated lncRNAs.

We initially focused on those lncRNAs most differentially expressed in breast cancers in comparison to benign adjacent tissue (Supplementary Data 1), utilizing a non-parametric differential expression tool for RNA-seq called Sample Set Enrichment Analysis (SSEA)[7]. After applying an expression filter (at least one fragments per kilobase of transcript per million mapped reads (FPKM) expression in the breast samples in the top 5% based on gene expression level), we identified 437 of the most differentially expressed lncRNAs in breast cancer (Supplementary Data 2). Interestingly, unsupervised hierarchical clustering of the samples based on expression of these lncRNAs across all breast cancer samples (Methods section) largely separated out the breast cancer samples by PAM50 subtypes[22,23], suggesting that lncRNAs may be contributing to the distinct biology of these subtypes (Fig. 1a). While lncRNA expression was unable to distinguish between the ER-driven luminal A and luminal B subtypes, the luminal subtypes were well separated from the HER2, basal and normal subtypes (Fig. 1a). In addition to separating out the clinical subtypes of breast cancer, the lncRNAs themselves separated into three distinct clusters. The first cluster (Fig. 1a, 'Luminal') contains lncRNAs overexpressed mostly in luminal A

and luminal B samples, with little expression in samples of the other subtypes, and little expression in normal samples. The next cluster contains lncRNAs upregulated across all breast cancer samples (Fig. 1a, 'Upregulated'), and this cluster included the known breast cancer lncRNA, HOTAIR. The third cluster (Fig. 1a, 'Downregulated') contains lncRNAs downregulated in breast cancers. The lncRNAs in the luminal cluster present a particularly intriguing class of potentially oestrogen-responsive lncRNAs.

Using the 947 breast tumour RNA-seq samples (Supplementary Data 1), we identified lncRNAs differentially expressed in ER-positive versus ER-negative breast tumours (Fig. 1b, Supplementary Data 2). As expected, the expression of lncRNAs differentially expressed in ER-positive tumours separated the luminal tumours from the basal and HER2 on unsupervised hierarchical clustering (Fig. 1b). Quite interestingly, a number of lncRNAs that were downregulated in ER-positive samples exhibited increased expression in the basal samples (Fig. 1b, 'Basal lncRNAs'). While these basal lncRNAs were identified in an ER-positive versus ER-negative cancer analysis, a number of them also exhibit low expression in normal breast tissue (Supplementary Fig. 1). Given that a paucity of known driver genes exist for basal breast cancers and that these tumours are the most clinically aggressive, these basal-specific lncRNAs may represent an exciting future area for basal breast cancer biology.

We set out to investigate potentially oncogenic ER-regulated lncRNAs by intersecting the lncRNAs upregulated in both the cancer versus normal (Fig. 1a) and ER-positive versus ER-negative (Fig. 1b) analyses. Sixty-three lncRNAs were upregulated in both the cancer versus normal analysis and the ER-positive versus ER-negative analysis (Supplementary Data 2, Fig. 1c). To prioritize the most biologically and clinically relevant lncRNAs, we focused on lncRNAs most highly expressed in breast cancer tissues, and those most directly regulated by ER, based on ER binding to the targets' promoter as well as the degree of induction of expression following oestrogen stimulation in breast cancer cells (Fig. 1c and Supplementary Fig. 2a). This approach nominated *DSCAM-AS1* as a lncRNA expressed at a very high level in breast cancer tissues, containing ER promoter binding, and exhibiting the strongest oestrogen induction in MCF7 and T47D cells by both RNA-seq and quantitative PCR (qPCR) validation (Fig. 1c and Supplementary Fig. 2a). We thus selected *DSCAM-AS1* for further investigation.

### Characterization of *DSCAM-AS1*.

*DSCAM-AS1* has been previously reported to be involved in the proliferation of a luminal breast cancer cell line[20]. It exhibits a highly cancer-specific expression pattern, mostly in breast cancer and lung adenocarcinoma, in transcriptome sequencing data from a cohort of 6,503 cancer and normal tissues and cell lines from the TCGA and the Michigan Center for Translation Pathology[7] (Fig. 2a). Supporting its association with ER biology, *DSCAM-AS1* expression is highly enriched (Student's *t*-test, $P$ value $< 10E^{-5}$) in ER-positive tumours among the breast cancer samples in this RNA-seq cohort with ER status determined by IHC (Fig. 2b and Supplementary Data 1). In addition, analysis of RNA-seq performed on 50 breast cancer cell lines[24] revealed that expression of *DSCAM-AS1* is highly specific to ER-positive cell lines (Fig. 2c and Supplementary Fig. 2b). Further supporting the association of ER with *DSCAM-AS1*, ER chromatin immunoprecipitation-sequencing (ChIP-seq) in both MCF7 and T47D identified ER binding to the *DSCAM-AS1* promoter following oestrogen stimulation (Fig. 2d), and this finding was confirmed by ChIP-qPCR of the *DSCAM-AS1* promoter

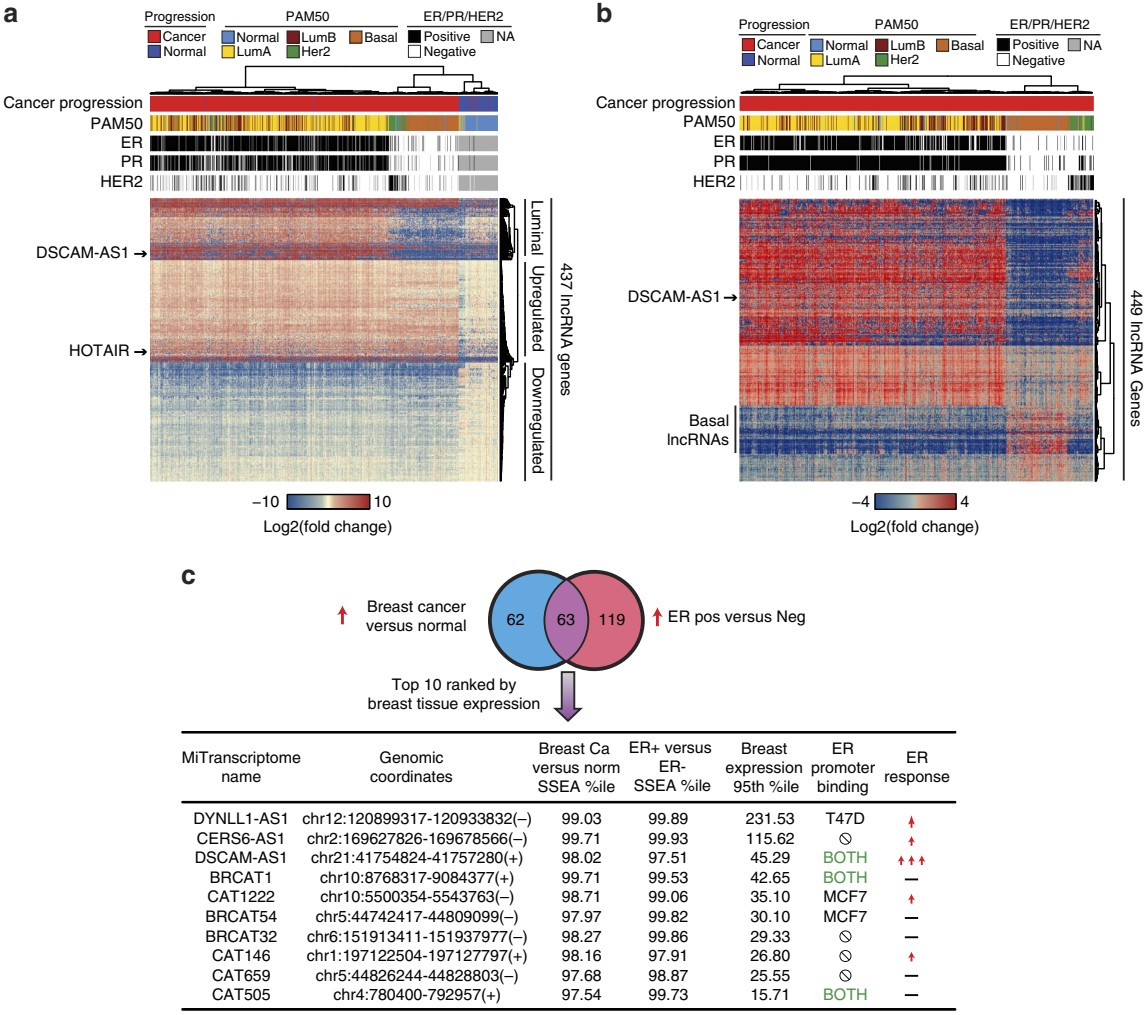

**Figure 1 | Identification of ER and breast cancer-associated lncRNAs.** (**a**) Heatmap depiction of the top cancer versus normal differentially expressed lncRNAs among the TCGA breast RNA-seq cohort ($n = 946$). 437 lncRNAs were differentially expressed with an SSEA FDR<1e-5 and an SSEA percentile cutoff of 0.975 (Methods section). Expression values are depicted as log2 of the fold change over the median of the normal samples ($n = 104$). Unsupervised hierarchical clustering was done on both lncRNAs and patients. Cancer progression, PAM50 classification, and ER, PR, and HER2 status are shown above heatmap. LncRNAs clustered into 3 distinct categories, 'Luminal', 'Upregulated', and 'Downregulated'. Two representative lncRNAs are highlighted. (**b**) Heatmap depiction of the top ER-positive versus ER-negative lncRNAs. 449 lncRNAs met the SSEA criteria described in **a**. Unsupervised clustering was performed for samples and lncRNAs. Expression values depicted as log2 of the fold change over the median of the ER-negative samples ($n = 538$). Cancer progression, PAM50 classification, and ER, PR and HER2 status are shown above heatmap. One representative lncRNA is highlighted along with a group of lncRNAs with basal-specific expression. (**c**) Venn diagram of the intersection of the breast cancer versus normal and ER-positive versus ER-negative analyses. Intersection is shown for the overexpressed lncRNAs in both categories. The top 10 lncRNAs based on expression level in breast cancer tissues (expression value of 95th percentile sample) are depicted in table. ER promoter binding determined via ChIP-seq is depicted (in either MCF7, T47D cell lines, or both) along with expression response from RNA-seq following 3 h of oestrogen stimulation in MCF7 cells (one arrow represents >1.5 fold increase, three arrows represents >2.5 fold increase).

(Supplementary Fig. 2c). The isoforms of *DSCAM-AS1* in MCF7 cells were identified using 3′ and 5′ RACE (Fig. 2d and Supplementary Table 1). *DSCAM-AS1* expression is induced in both MCF7 and T47D cells after oestrogen stimulation, and this induction is reversed with addition of tamoxifen, corroborating that ER is in fact regulating the expression of this lncRNA (Fig. 2e). Expression of known ER-regulated protein-coding genes *GREB1* and *PGR* follow the same pattern of response to oestrogen, while the lncRNA *MALAT1*, serving as a negative control, is not induced by oestrogen (Fig. 2e). In addition to being oestrogen-responsive, *DSCAM-AS1* expression is present in both the cytoplasm and nucleus at nearly identical fractions in both MCF7 and T47D cells (Supplementary Fig. 2d), and the identity of *DSCAM-AS1* as a noncoding gene was corroborated

using the CPAT tool[25] (Supplementary Fig. 2e). We used single-molecule fluorescence *in situ* hybridization (ISH) to further dissect the subcellular localization and gene expression levels of DSCSM-AS1 in breast cancer cells. To this end, we designed probes that targeted all potential isoforms of the transcript predicted by RACE. On staining, we found that each MCF7 cell expressed ~800 copies of the DSCAM-AS1 transcript, almost half as much as the expression level of GAPDH (Supplementary Fig. 2f,g), additionally the similar nuclear and cytoplasmic localization was corroborated by ISH (Supplementary Fig. 2h). While the abundance of DSCAM-AS1 was lower in T47D cells (~260 molecules per cell, Supplementary Fig. 2i,j), the relative expression level (compared with GAPDH) and the subcellular localization pattern were very

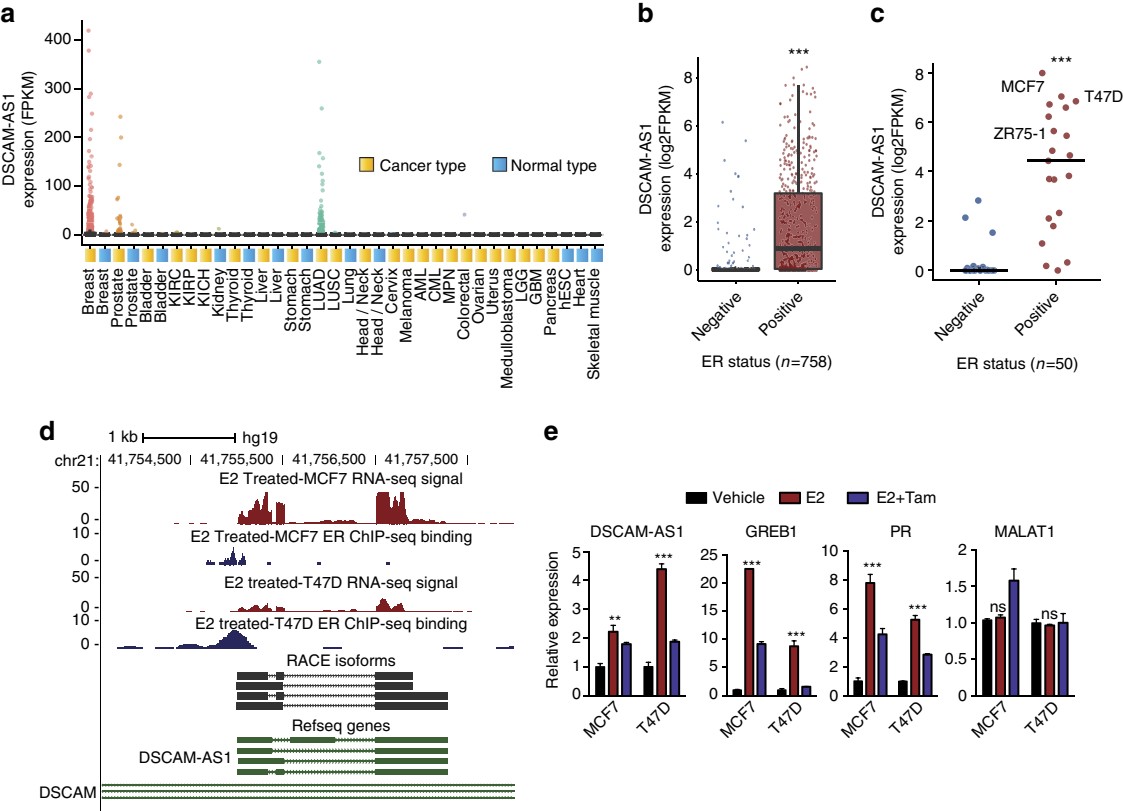

**Figure 2 | Characterization of *DSCAM-AS1*.** (**a**) Plot highlighting the expression in FPKM of *DSCAM-AS1* in the 6,503 sample MiTranscriptome RNA-seq compendium[7] categorized by the different cancer/tissue types. Each point represents one RNA-seq tissue sample. (**b**) Expression of *DSCAM-AS1* is significantly higher in ER-positive breast cancer tissue samples ($n = 584$) compared with ER-negative samples ($n = 174$). Expression was analysed in samples for which ER IHC was performed. Each point represents one RNA-seq sample. ***$P < 0.0001$, comparing ER-positive with -negative. (**c**) Expression of *DSCAM-AS1* by RNA-seq in breast cancer cell lines categorized by ER status. *DSCAM-AS1* expression is significantly higher in ER-positive cell lines ($n = 21$) versus ER-negative cell lines ($n = 29$). Each point represents one cell line. ***$P < 0.0001$, comparing ER-positive to –negative via Student's *t*-test. (**d**) UCSC genome browser depiction of *DSCAM-AS1* region on chromosome 21. RNA-seq expression track shown in red, and ER ChIP-seq shown in blue. Refseq transcripts shown in green. RACE verified transcript structure shown in black. (**e**) qPCR expression of *DSCAM-AS1*, *GREB1*, *PGR*, and *MALAT1* 8 h following addition of DMSO vehicle (black), 10 nM estrogen (red), and 10 nM estrogen and 1 μM tamoxifen (blue) in MCF7 and T47D cell lines. Error bars represent s.e.m. for three biological replicates. **$P < 0.001$, ***$P < 0.0001$, NS: $P > 0.01$ comparing with vehicle for each condition via Student's *t*-test. NS, not significant.

similar to those observed in MCF7 cells (Supplementary Fig. 2k).

**DSCAM-AS1 is implicated in cancer aggression**. We next investigated the clinical relevance of *DSCAM-AS1*. Given that *DSCAM-AS1* is a lncRNA, its expression is not measured by most traditionally used microarrays, which are the primary high-throughput platforms annotated with reliable clinical outcomes in breast cancer[26]. As a surrogate, we employed a guilt-by-association analysis to interrogate the clinical relevance of those genes most correlated to *DSCAM-AS1*. Given that *DSCAM-AS1* is an ER-regulated lncRNA, correlation was performed using only ER-positive breast cancers, to ascertain clinical relevance in the breast cancer samples in which *DSCAM-AS1* would be enriched and most relevant. We obtained a number of breast cancer clinical data sets from Oncomine[27] containing gene expression sets associated with the presence of cancer (versus normal tissue), high clinical grade, recurrence, survival and metastasis[22,23,26–40] (Methods section). We assessed for the overlap between these gene sets with the genes most positively or negatively correlated to *DSCAM-AS1*. *DSCAM-AS1* positively correlated genes were significantly associated with clinical signatures associated with increased cancer aggression, tamoxifen resistance, higher grade,

stage and metastasis (Fig. 3a,b, Supplementary Data 3 and 4). Similarly, the *DSCAM-AS1* negatively correlated genes associated with clinical signatures that portended a more favourable clinical outcome (Supplementary Fig. 3a,b, Supplementary Data 3 and 4). For many of the clinical concepts, *DSCAM-AS1* positively correlated genes displayed a clinical association comparable to those genes most correlated to *EZH2*, a gene known to be a marker of clinical aggressiveness in breast cancer[41], while genes correlated to other lncRNAs expressed in breast tissue, such as *HOTAIR*, *MALAT1* and *NEAT1*, showed modest-to-no association (Fig. 3b, Supplementary Fig. 3b, Supplementary Data 3 and 4). In addition, performing a Gene Set Enrichment Analysis (GSEA)[42] on all genes correlated to *DSCAM-AS1* yielded significant association with a myriad of breast cancer, cancer aggressiveness, and ER- and tamoxifen-associated gene signatures (Supplementary Fig. 3c). While ER-positive breast cancers typically result in better clinical outcomes[23], among the luminal breast cancers, *DSCAM-AS1* is expressed significantly higher in luminal B, a clinical subtype containing most of the clinically aggressive ER-positive breast cancers[22,23] (Fig. 3c). Despite these associations of clinical aggression with *DSCAM-AS1*, in a survival analysis of the ER-positive TCGA breast samples, expression of *DSCAM-AS1* was not significantly associated with clinical outcome (Supplementary Fig. 3d). Definitive assessment of

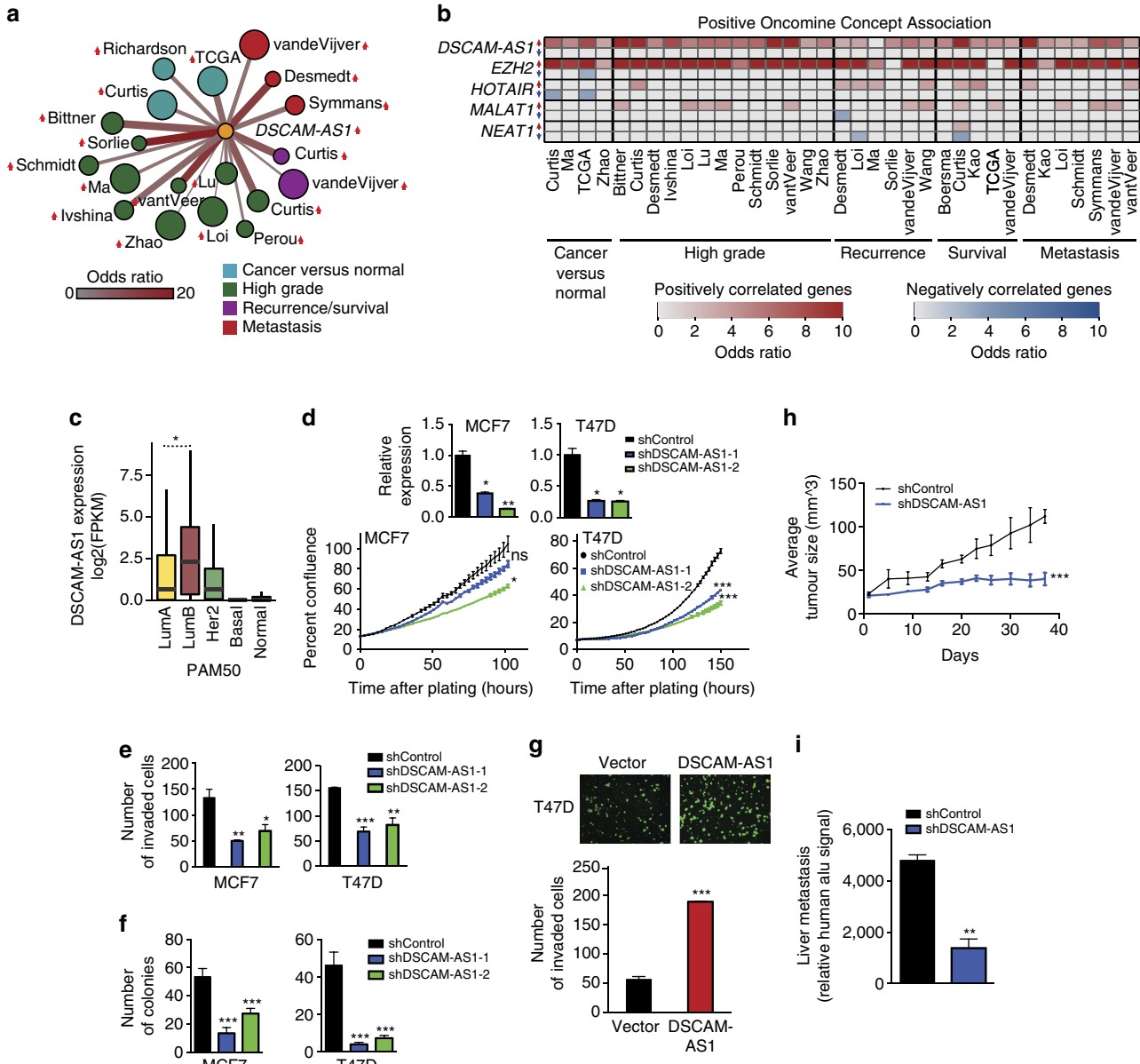

**Figure 3 | *DSCAM-AS1* is implicated in cancer aggression clinically and in cell lines.** (**a**) Cytoscape depiction of the overlap between the 150 genes most positively correlated with *DSCAM-AS1* and clinical signatures from Oncomine[27] for breast cancer clinical outcomes (i.e., recurrence, survival and metastasis), high cancer grade, and cancer versus normal. All significant associations with an odds ratio >6 are shown (Fisher's *P* value <1E$^{-4}$). Size of node reflects the size of the gene signature, and the thickness/redness of the line represents the magnitude of the odds ratio. (**b**) Heatmap displaying the overlap between the top 150 genes correlated to *DSCAM-AS1*, *EZH2*, *HOTAIR*, *MALAT1* and *NEAT1* and the genes positively associated with various breast cancer clinical signatures (see above). For each gene, the top row depicts the odds ratio for the positively correlated genes (red), and the bottom row represents the odds ratio for the negatively correlated genes (blue). The first name of the author for each clinical study is listed. (**c**) Expression of *DSCAM-AS1* from breast cancer RNA-seq by PAM50 classification (n = 946). Luminal B expression is significantly greater than Luminal A (Student's *t*-test, *P* = 0.006) (**d**) Incucyte proliferation assay performed following knockdown of *DSCAM-AS1* using two independent shRNAs. Degree of knockdown determined by qPCR shown above. Error bars represent the s.e.m. for three biological replicates. *P < 0.01, **P < 0.001, ***P < 0.0001, NS: P > 0.01 comparing to shControl for each condition via Student's *t*-test. (**e**) Invasion assay following shRNA knockdown of *DSCAM-AS1* using Matrigel coated Boyden chamber assay. Error bars represent the s.e.m. for three biological replicates. *P < 0.01, **P < 0.001, ***P < 0.0001 comparing to shControl for each condition via Student's *t*-test. (**f**) Soft agar colony formation assay following shRNA knockdown of *DSCAM-AS1*. Error bars represent the s.e.m. for three biological replicates. ***P < 0.0001 comparing to shControl for each condition via Student's *t*-test. (**g**) Invasion assay following overexpression of *DSCAM-AS1* and *LacZ* control. Error bars represent the s.e.m. for three biological replicates. Representative invasion images shown above. ***P < 0.0001, comparing to vector overexpression via Student's *t*-test. (**h**) Mouse xenograft study of tumour growth for T47D cells with shRNA knockdown of *DSCAM-AS1*. Xenografts with shRNA knockdown of *DSCAM-AS1* (n = 24) exhibited reduced growth when compared to control shRNA knockdown (n = 26). Error bars represent the s.e.m. for all xenografts used. ***P < 0.0001, comparing to shControl via Student's *t*-test. (**i**) Assessment of xenograft metastasis to liver by human Alu PCR, which detects the human cancer cells in the mouse liver. Error bars represent the s.e.m. for three biological replicates. **P < 0.001 comparing with shControl via Student's *t*-test. NS, not significant.

survival in this cohort, however, will likely require more robust and longer-term clinical curation of the TCGA breast samples.

We then studied the role of *DSCAM-AS1* on oncogenic phenotypes in ER-positive breast cancer cell lines. In MCF7 and T47D cells, stable knockdown of *DSCAM-AS1* was achieved using shRNA approaches. *DSCAM-AS1* knockdown reduced the proliferative ability of both cell lines (Fig. 3d), diminished the ability of these cells to invade in a Boyden chamber invasion assay (Fig. 3e), and substantially abolished the ability of these cells to form colonies in soft agar (Fig. 3f). While ER regulates levels of *DSCAM-AS1*, ER expression and protein levels are not dependent on level of *DSCAM-AS1* (Supplementary Fig. 4a), ruling out the possibility that the phenotype observed could be explained through changes in the level of *ER*. In addition, knockdown of *DSCAM-AS1* exhibited no affect on RNA or protein levels of the *DSCAM* gene, in which *DSCAM-AS1* resides antisense and intronic (Supplementary Fig. 4b). To further demonstrate the impact of *DSCAM-AS1* on aggressive cancer phenotypes, we overexpressed *DSCAM-AS1* in T47D (Supplementary Fig. 4c) and ZR75-1 (Supplementary Fig. 4d), two ER-positive breast cancer cell lines with moderate *DSCAM-AS1* expression (Fig. 2c), and observed an increase in the invasion phenotype (Fig. 3g and Supplementary Fig. 4e). MCF7 cells were not included in the overexpression studies as *DSCAM-AS1* is already expressed at a very high level in these cells (Fig. 2c). Overexpression was also tested in MDA-MB-231 cells (Supplementary Fig. 4f), a common ER-negative cell line. However, exogenous *DSCAM-AS1* was unable to confer oncogenicity via proliferation (Supplementary Fig. 4g) and invasion (Supplementary Fig. 4h). This phenomenon may be explained by a requisite genetic and epigenetic milieu provided by ER-positive cells in order for *DSCAM-AS1* to confer its cancer phenotype, and more investigation into the precise mechanisms through which it acts will shed light on this finding. Furthermore, the simple presence of *DSCAM-AS1* alone is not sufficient to make cells highly aggressive, as evidenced by its high expression in ER-positive cell lines that are moderately invasive (for example, MCF7). To further characterize the impact of *DSCAM-AS1* on cancer phenotype, we performed a mouse xenograft tumour growth assay, showing that loss of *DSCAM-AS1* reduces the growth of implanted T47D cells *in vivo* (Fig. 3h). The metastatic potential of these implanted cells were also reduced with *DSCAM-AS1* knockdown, as evidenced through decreased liver metastasis following xenograft (Fig. 3i).

**Role of hnRNPL in *DSCAM-AS1* mechanism.** LncRNAs have been shown to be functional through their binding interactions with other RNAs, DNA, and with proteins[2]. Thus, identifying protein binding partners for *DSCAM-AS1* is a crucial step in determining the mechanism through which it confers oncogenicity. To identify *DSCAM-AS1* binding partners, we performed pull-down of *DSCAM-AS1* and performed mass spectrometry on the pull-down product to identify proteins bound to *DSCAM-AS1* (Fig. 4a). The protein hnRNPL was observed to have the highest spectral counts for the sense form of *DSCAM-AS1* with zero spectral counts in the antisense pull-down (Fig. 4b). In addition, PCBP2, a protein known to complex with hnRNPL[43], was also among the top proteins bound to *DSCAM-AS1*. We thus investigated the interaction between *DSCAM-AS1* and hnRNPL further. HnRNPL is a protein widely expressed in many tissue types (Supplementary Fig. 5a) and has been implicated in regulating RNA stability and processing with subsequent effects on gene expression[44–47]. The binding of hnRNPL to *DSCAM-AS1* was confirmed by RNA pull-down followed by western blot, with no binding of hnRNPL to the negative control antisense transcript (Fig. 4c).

Other RNA-binding proteins did not bind *DSCAM-AS1*, however, suggesting that *DSCAM-AS1* does not promiscuously bind to RNA-binding proteins in general (Fig. 4c). To further confirm this binding interaction and its specificity, RNA immunoprecipitation (RIP) was performed with using antibodies directed against hnRNPL. *DSCAM-AS1* was highly enriched by anti-hnRNPL RIP in both MCF7 and T47D cells, while control coding and noncoding genes exhibited modest binding (Fig. 4d). In addition, anti-snRNP70 and anti-HuR RIP failed to pull-down *DSCAM-AS1*, further suggesting the specificity of the *DSCAM-AS1*-hnRNPL interaction (Supplementary Fig. 5b).

To more specifically investigate the functional relationship of *DSCAM-AS1* and hnRNPL, we performed rescue studies assessing the impact of *hnRNPL* knockdown on the invasive advantage conferred by *DSCAM-AS1* overexpression, observing that reduction of *hnRNPL* levels entirely reversed the increase in invasion observed on *DSCAM-AS1* overexpression (Fig. 4e, Supplementary Fig. 6a). Because there was only slight, non-significant reduction in invasion with *hnRNPL* knockdown in control cells, the marked reduction in invasion observed in the *DSCAM-AS1* overexpressing cells with *hnRNPL* knockdown may be the result of *hnRNPL* affecting invasion in a mechanism exclusive to *DSCAM-AS1*. So, to further characterize the functional relationship between *DSCAM-AS1* and hnRNPL, we set out to localize the binding site of hnRNPL within the *DSCAM-AS1* lncRNA. Using *in silico* prediction drawing from prior studies of hnRNPL crosslinking-immunoprecipitation sequencing (CLIP-seq)[48], a single strong predicted binding peak was identified near the 3'-end of *DSCAM-AS1* (Fig. 4f). HnRNPL has been shown to bind CACA-rich RNA sites[45], and the predicted binding region possessed a 10 base pair CACA stretch. To identify if this predicted region does in fact account for the hnRNPL binding, multiple mutant forms of *DSCAM-AS1* were created with or without the binding site. *DSCAM-AS1-5* and *DSCAM-AS1-3* are large deletion mutants containing only the 5'- and 3'-end, respectively, with only *DSCAM-AS1-3* possessing the predicted binding site, and *DSCAM-AS1-D* is a mutant form with the 27 nucleotides comprising the predicted binding site deleted (Fig. 4f,g, red). The various mutant forms of *DSCAM-AS1* were expressed in HEK293, a cell line that lacks endogenous *DSCAM-AS1* expression while still expressing hnRNPL (Supplementary Fig. 6b). While both the full-length and *DSCAM-AS1-3* mutant retained hnRNPL binding, loss of the predicted binding region was effective in abrogating hnRNPL binding via both Western blot following RNA pull-down (Fig. 4g) and by qPCR following hnRNPL RIP (Supplementary Fig. 6c). All deletion mutants were expressed at comparable levels, ruling out the possibility of falsely diminished binding due to failed expression of the mutant construct (Supplementary Fig. 6d). RNA secondary structure is a crucial component of RNA functionality and is a key player in RNA-protein interactions. While the 27 nucleotide deletion in the *DSCAM-AS1-D* mutant is a small fraction of the total number of bases in the transcript, to ensure that this deletion was not causing a marked RNA secondary structure change, we investigated the impact of this deletion on RNA secondary structure via the RNAfold structure prediction tool[49]. Evidenced by a minimal free energy prediction, the posited secondary structure of *DSCAM-AS1* is largely similar to that of *DSCAM-AS1-D* (Supplementary Fig. 6e), suggesting that the loss of hnRNPL binding observed with the *DSCAM-AS1-D* mutant is not due to a dramatic secondary structure rearrangement. Quite interestingly, overexpression of the *DSCAM-AS1-D* mutant in T47D cells failed to recapitulate the increased invasion observed when overexpressing full-length *DSCAM-AS1* (Fig. 4h). This finding, in combination with the rescue studies following *hnRNPL* knockdown (Fig. 4e), strongly

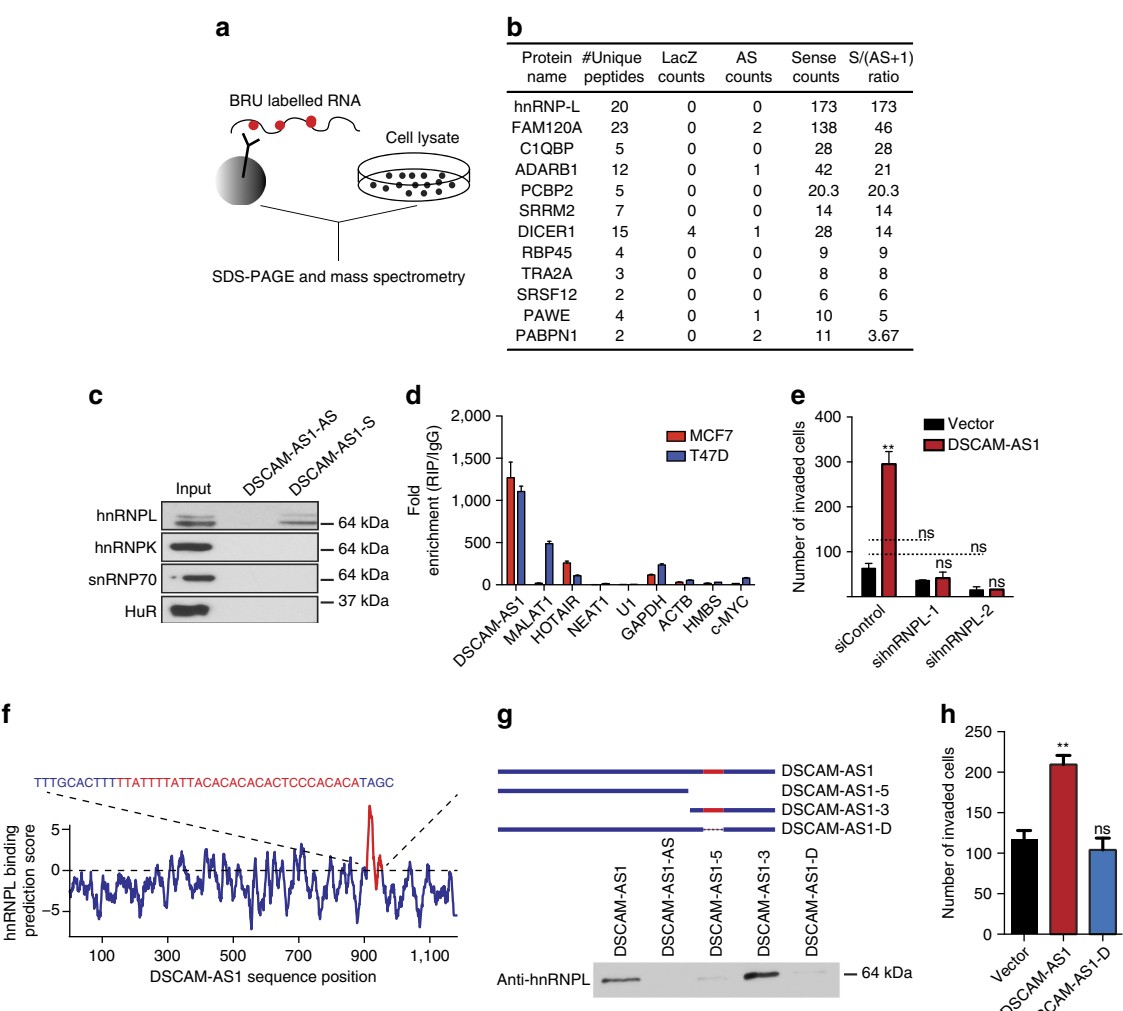

**Figure 4 | Physical and functional relationship of *DSCAM-AS1* with hnRNPL.** (**a**) Schematic representation of the RNA pull-down technique used to identify protein binding partners of *DSCAM-AS1*. The BRU lableled RNA transcripts are incubated with cell lysate from T47D cells and the eluted protein is resolved by SDS–PAGE. RNA-bound protein product is then processed by mass spectrometry. (**b**) Top protein binding partners for *DSCAM-AS1*. Pull-down of *LacZ* and antisense *DSCAM-AS1* used as control. S/AS ratio determined as sense counts divided by 1 + antisense counts. (**c**) Western blot of hnRNPL, hnRNPK, snRNP70, and HuR following pull-down of BRU labelled *DSCAM-AS1* and antisense *DSCAM-AS1*. (**d**) qPCR following RIP for hnRNPL performed in MCF and T47D cells. Data represented as fold-enrichment over IgG RIP. Error bars represent the s.e.m. for three biological replicates. (**e**) Invasion assay for T47D cells overexpressing *LacZ* control or *DSCAM-AS1* following siRNA-mediated knockdown of *hnRNPL*. Error bars represent the s.e.m. for two biological replicates. **$P < 0.001$, NS: $P > 0.01$ comparing to siControl (unless otherwise specified with dotted line) for each condition via Student's *t*-test. (**f**) Per base *in silico* prediction for binding of hnRNPL to *DSCAM-AS1*. One strong predicted binding peak exists in the 3′ region of *DSCAM-AS1* shown in red. (**g**) Schematic depicting the mutant forms of *DSCAM-AS1* generated with or without the predicted binding site (top). Western blot for hnRNPL shown following pull-down of each mutant form of *DSCAM-AS1* in HEK293 cells (bottom). (**h**) Invasion assay in T47D cells overexpressing *LacZ* control, full-length *DSCAM-AS1*, and the *DSCAM-AS1-D* mutant. Error bars represent the s.e.m. for three biological replicates. **$P < 0.001$, NS: $P > 0.01$ comparing to vector overexpression for each condition via Student's *t*-test. NS, not significant; siRNA, small interfering RNA.

suggest that *DSCAM-AS1* promotes oncogenicity via its interaction with hnRNPL in these ER-positive breast cancer cells.

**Role of *DSCAM-AS1* in tamoxifen resistance.** A substantial number of patients with ER-positive breast cancer eventually develop resistance to endocrine therapy and present with clinical recurrence and metastasis[11,50,51]. Thus, as *DSCAM-AS1* is implicated in poor-prognosis ER-positive breast cancer (Fig. 3a-c and Supplementary Fig. 3), we set out to investigate its potential role in subverting oestrogen dependence and promoting resistance to anti-oestrogen therapies. We continuously passaged MCF7 cells in 1 uM tamoxifen for 6 months until we attained a subpopulation of MCF7 cells that were able to grow in

in tamoxifen and termed these tamoxifen-resistant MCF7 cells (TamR-MCF7). Interestingly, although expression of canonical ER targets (*GREB1* and *PGR*) was decreased compared to the parental MCF7 cells, *DSCAM-AS1* expression was significantly upregulated despite already being expressed at very high levels in MCF7 cells (Fig. 5a). The levels of *ER* were also increased, which is likely a compensatory upregulation in response to the continual anti-oestrogen effects of tamoxifen. Additionally, short-term tamoxifen treatment of parental MCF7 cells transiently reduced *DSCAM-AS1* levels at 8hrs following tamoxifen treatment, with a rise back to pre-treatment levels after 24 h (Supplementary Fig. 7a). In contrast, canonical ER target, *GREB1*, exhibited pronounced expression reduction at both the short- and long-term timescale (Supplementary Fig. 7b). To interrogate whether

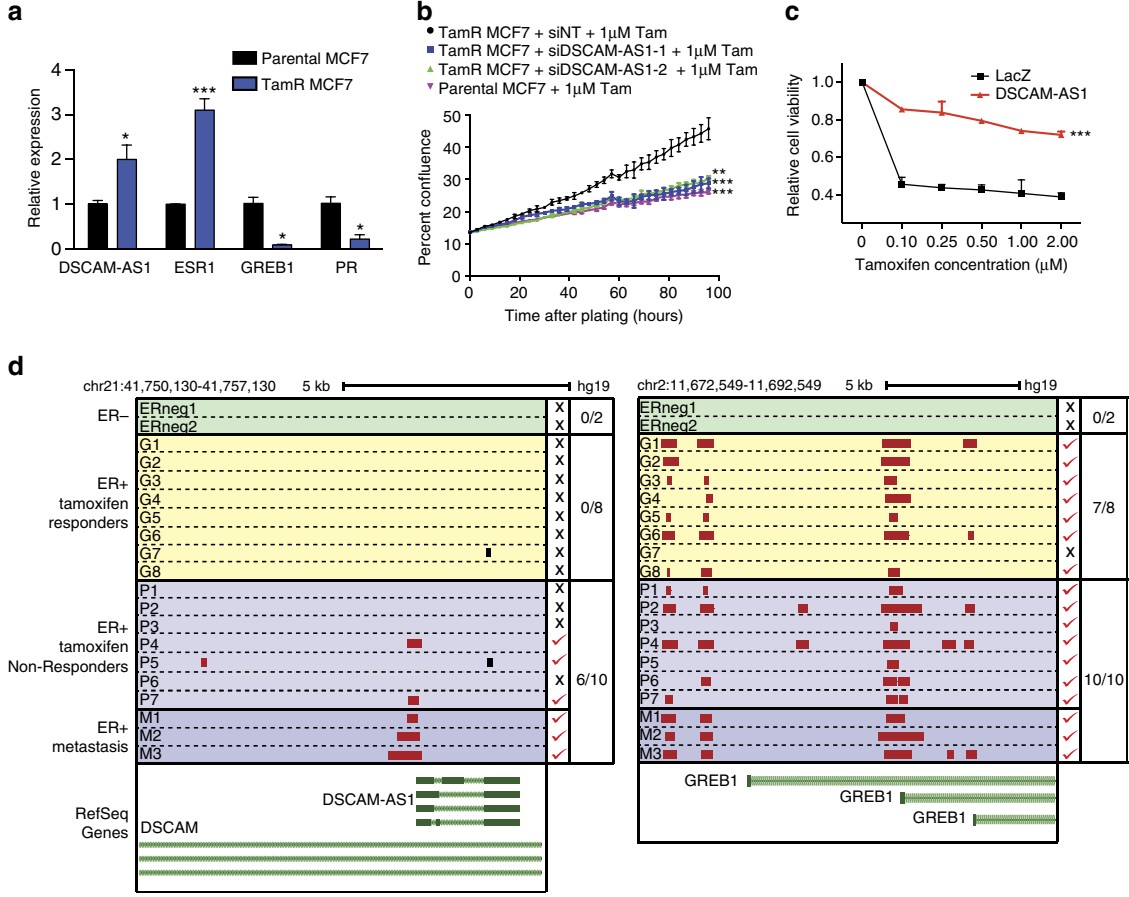

**Figure 5 | *DSCAM-AS1* is implicated in tamoxifen resistance.** (**a**) qPCR expression of *DSCAM-AS1*, *ESR1*, *GREB1* and *PGR* in tamoxifen-resistant MCF7 cells relative to parental MCF7. Error bars represent the s.e.m. for three biological replicates. *$P<0.01$, ***$P<0.0001$, comparing to parental MCF7 for each condition via Student's *t*-test. (**b**) Proliferation assay in parental MCF7 cells and in TamR-MCF7 cells following siRNA-mediated knockdown of *DSCAM-AS1* via two independent siRNAs. Error bars represent the s.e.m. for three biological replicates. **$P<0.001$, ***$P<0.0001$, comparing to parental TamR siNT for each condition via Student's *t*-test. (**c**) WST viability assay following 10 days of culture in varying levels of tamoxifen performed for T47D cells overexpressing LacZ control and *DSCAM-AS1*. Error bars represent the s.e.m. for three biological replicates. ***$P<0.0001$, comparing to LacZ overexpression via Student's *t*-test. (**d**) Depiction of oestrogen receptor binding to the *DSCAM-AS1* (left) and *GREB1* (right) promoters via ChIP-seq performed in primary and metastatic breast cancer tumour tissues. ER status and response to tamoxifen treatment detailed to left. ER binding peaks (determined using MACS software) are depicted in red (for promoter binding) and black (for non-promoter binding). Promoter defined as 5KB upstream of any transcriptional start site. ER promoter binding indicated by red check or 'x' to the right. Genomic coordinates in hg19 listed above. siRNA, small interfering RNA.

this upregulation of *DSCAM-AS1* in the TamR-MCF7 cells is functionally significant, we assessed the proliferative capacity of these cells following *DSCAM-AS1* knockdown. With knockdown levels of *DSCAM-AS1* comparable to the endogenous levels in parental MCF7 cells (Supplementary Fig. 7c), knockdown of *DSCAM-AS1* in TamR-MCF7 cells led to a loss of their baseline proliferative advantage when cultured in tamoxifen, exhibiting a proliferation profile nearly identical to that of the parental MCF7 cells (Fig. 5b). Additionally, knockdown of *hnRNPL* in these cells produced a similar loss of proliferative capacity in the TamR-MCF7 cells (Supplementary Fig. 7d,e), suggesting that both *DSCAM-AS1* and hnRNPL may be playing a role in promotion of the tamoxifen resistance developed by these cells.

We also interrogated the ability of *DSCAM-AS1* to confer tamoxifen resistance in native T47D cells via overexpression of *DSCAM-AS1*. *DSCAM-AS1* overexpression was also associated with tamoxifen-resistant growth in a dose-dependent manner, with a striking increase in cell viability at levels of tamoxifen as low was 100 nM (Fig. 5c). Additionally, in line with the ability of *DSCAM-AS1* to provide oestrogen-independent growth advantage, cells overexpressing *DSCAM-AS1* also exhibited a

proliferative advantage when grown in oestrogen-deprived medium compared to normal serum (Supplementary Fig. 7f). This growth advantage was abolished with the addition of oestrogen, and returned with the subsequent addition of tamoxifen (Supplementary Fig. 7f). Conversely corroborating the relationship of *DSCAM-AS1* on oestrogen dependence in these cells, we witnessed an increased oestrogen dependence of T47D cells following *DSCAM-AS1* knockdown (Supplementary Fig. 7g).

To corroborate our *in vitro* findings in a tissue model, we obtained data previously generated performing ChIP-seq for ER in primary and metastatic breast tumour tissue[13]. These tumours were grouped into the following categories as previously described[13]: primary ER-negative ($n=2$), primary ER-positive, tamoxifen-responder ($n=8$), primary ER-positive, tamoxifen-non-responder ($n=7$), metastatic ER-positive ($n=3$). Strikingly, investigation of the *DSCAM-AS1* promoter revealed that ER preferentially binds to the *DSCAM-AS1* promoter in tumours with clinical aggression (ie, metastatic and tamoxifen non-responders; Fig. 5d), while a canonical ER target, *GREB1*, exhibits ER-biding to its promoter in nearly all ER-positive tumours, lacking preference for the more clinically aggressive

tumours. Altogether, these data suggest that the association between *DSCAM-AS1* expression with clinical aggressiveness in ER-positive breast cancer samples may be explained, in part, by the ability of *DSCAM-AS1* to facilitate oestrogen-independent oncogenicity, thus potentially promoting resistance to endocrine therapy with tamoxifen.

## Discussion

Further investigation and study of the mechanisms through which ER-dominant breast cancers become aggressive and eventually evade traditional clinical therapies is of intense clinical interest. In this study, we identify a myriad of potentially ER-associated lncRNAs, and functionally and mechanistically characterize one of the most intriguing candidates. Nevertheless, further investigation of some of these other lncRNAs may also contribute to our understanding of ER biology and ER-driven oncogenesis. LncRNAs have been shown to function through multiple mechanisms, and the study of the interaction of *DSCAM-AS1* with hnRNPL is a promising step towards understanding the ways through which this molecule executes its oncogenic function. While we show that the binding of hnRNPL to *DSCAM-AS1* is responsible for at least some of its oncogenicity, a further understanding of how the interaction between hnRNPL and *DSCAM-AS1* is mediating this phenotype is necessary.

Novel mediators of tumour aggression, such as *DSCAM-AS1*, can provide insight into the mechanism of endocrine therapy resistance. This increased understanding may in turn lead to more effective strategies to overcome this resistance, which is one of the last, great clinical challenges in treatment of ER-positive breast cancer. In addition, there is little known regarding the role of noncoding RNAs in developing resistance to anti-oestrogen therapy, with a small number of studies implicating some of the more prominent, well characterized breast cancer lncRNAs[52,53]. *DSCAM-AS1* is just one of many potentially relevant ER-regulated lncRNAs in breast cancer, and further investigation of the other candidates is likely to yield a greater understanding of ER-mediated cancer biology. Ultimately, this study provides key insight into the role of lncRNAs in ER breast cancer biology, and is an important step in better understanding this common disease.

## Methods

**Cell lines and cell culture.** All cell lines were obtained from the American Type Culture Collection (Manassas, VA). Cell lines were maintained using standard media and conditions. Specifically, T47D cells were maintained in Roswell Park Memorial Institute (RPMI) 1640 medium supplemented with 10% fetal bovine serum, 1% penicillin-streptomycin and $5 \, mg \, ml^{-1}$ insulin. ZR75-1 cells were maintained in RPMI 1640 medium supplemented with 10% fetal bovine serum, 1% penicillin-streptomycin. HEK293 cells were maintained in DMEM plus 10% fetal bovine serum (FBS) plus 1% penicillin-streptomycin. MCF7 cells were cultured in Dulbecco's modified Eagle's medium plus GlutaMAX (DMEM, Invitrogen) containing 10% fetal bovine serum (Hyclone) and 1% penicillin-streptomycin. To establish the tamoxifen-resistant cell line, MCF7 cells were grown in IMEM phenol-red free medium with 10% Charcoal-stripped FBS in the presence of 1 uM (Z)-4-hydroxytamoxifen (Sigma) for 6 months. All cell lines were grown at 37 °C in a 5% $CO_2$ cell culture incubator, and were genotyped for identity at the University of Michigan Sequencing Core and tested routinely for *Mycoplasma* contamination.

**Cell proliferation assay.** Cells were seeded in a 48-well plate at $3 \times 10^4$ cells per well. Plates were added to Incucyte machine (Essen Bioscience) 16–20 h following seeding. Growth curves were constructed by imaging plates using the Incucyte system, where the growth curves are generated from confluence measurements acquired during continuous kinetic imaging. Four wells were measured per condition. For tamoxifen treatment, 16–20 h after seeding, the medium was changed to RPMI phenol-red free medium containing 10% charcoal-treated FBS in the presence of 1 uM tamoxifen or ethanol. Growth curves were obtained using Incucyte system as described above.

**Cell viability assay.** Cells were seeded in 96-well plates at 5,000 cells per well in a total volume of 100 μl media containing 10% FBS. Serially diluted tamoxifen in 100 μl of media was added to the cells 12 h after seeding. Medium containing tamoxifen was replenished every 2–3 days. Following 10 days of incubation, cell viability was assessed by WST assay (WST-8, Dojindo). All assays were performed in triplicate and repeated at least three times. The relative cell viability was expressed as a percentage of the control that was treated with vehicle solutions.

**Soft agar colony formation assay.** 10,000 cells were suspended in medium containing 0.3% agar, 10% FBS, and layered on medium containing 0.6% agar and 10% FBS in six-well plate. Colonies were stained for 18–24 h with iodonitrotetrozolium chloride (Sigma #18377) following 3 weeks of incubation. Colonies from three replicate wells were quantified.

**Quantitative RT–PCR assay.** The miRNeasy mini kit was utilized to isolate RNA from cell lysates. From 1 μg of isolated RNA, SuperScript III (Invitrogen) and Random Primers (Invitrogen) were used to generate cDNA according to the manufacturer's protocol. The ABI7900 HT Fast Real time system (Applied Biosystems) was utilized for quantitiative reverse transcriptase–PCR (qRT–PCR) reactions. Gene-specific primer were designed using the Primer3 software and were subsequently synthesized by IDT Technologies. A relative quantification method was used in analysing the qRT–PCR data and data were depicted as average fold change versus the control (as internal reference, GAPDH and actin were utilized). All primers used for qPCR are detailed in Supplementary Table 2. Three technical replicates were used in each assay, and all data shown was performed with at least three biological replicates.

**Oestrogen and tamoxifen treatment.** To evaluate the effect of oestrogen stimulation, cells were first hormone depleted via growth in phenol-red free medium containing 10% charcoal-treated FBS for 72 h and then treated with ethanol vehicle, 10 nM β-estradiol, or 10 nM β-estradiol plus 1 uM tamoxifen. After 10 h, RNA was isolated as described above and qPCR was performed as described above using Power SYBR Green Mastermix (Applied Biosystems).

**Subcellular fractionation.** Cellular fractionation was performed using a RiboTrap Kit (MBL International), according to manufacturer's instructions. RNA was isolated and qRT–PCR was performed as described above.

**Knockdown and overexpression studies.** Knockdown of *DSCAM-AS1* and *hnRNPL* in T47D and MCF7 cells was accomplished by small interfering RNA from Dharmacon. Transfections were performed with OptiMEM (Invitrogen) and RNAi Max (Invitrogen) per manufacturer instruction. Target sequences used for shRNA or small interfering RNA knockdown are listed in Supplementary Table 3. For stable knockdown of *DSCAM-AS1*, MCF7 and T47D cells were transfected with lentiviral constructs containing 2 different *DSCAM-AS1* shRNAs or no targeting shRNAs in the presence of polybrene ($8 \, \mu g \, ml^{-1}$ Supplementary Table 3). After 48 h, transduced cells were grown in culture media containing $2 \, \mu g \, ml^{-1}$ puromycin.

For *DSCAM-AS1* overexpression, the predominant isoform (isoform 2, Supplementary Table 1) was cloned into the pLenti6.3 vector (Invitrogen) using PCR8 non-directional Gateway cloning (Invitrogen) as an initial cloning vector and shuttling was then done to pLenti6.3 using LR clonase II (Invitrogen) according to the manufacturer's instructions. As control, *LacZ* was also cloned into the same vector system. The primers for making *DSCAM-AS1* mutation and truncations are listed in Supplementary Table 2. Lentiviral particles were made and T47D and ZR75.1 cells were transduced as described above. Stable cell lines were generated by selection with $3 \, \mu g \, ml^{-1}$ blasticidin. Transient transfection of *DSCAM-AS1* and its derivative mutants was done in HEK293 cells was performed with Lipofectamine LTX (Invitrogen) following manufacturer's instruction. Cells were collected at 48 h post transfection.

***In vitro* RNA-binding assay.** The RNA-binding assay was performed according to the protocol of the RiboTrap Kit (MBL International). In brief, 5-bromo-UTP (BrU) was randomly incorporated into sense *DSCAM-AS1*, antisense *DSCAM-AS1*, and *LacZ* control via PCR-based transcription. The primers are shown in Supplementary Table 2. The the BrU labelled RNA transcripts were bound to beads conjugated with anti-BrdU antibodies. Then, the cytoplasmic or nuclear fractions from MCF7 or T47D cells were mixed for 2 h. Samples were washed four times with Wash Buffer II before elution. The samples were sent to the Michigan Center for Translational Pathology proteomic core facility for mass spectrometry.

**Mass spectrometry.** The samples were treated with SDS-PAGE loading buffer supplied with 10 mM DTT for 5 min at 85 °C. The proteins were alkylated by the addition of iodoacetamide to the final concentration of 15 mM. The samples were subjected to SDS–PAGE and the whole lanes were cut out and digested with trypsin in-gel for 2 h. The resulting peptides were extracted, dried and resuspended in 0.1% formic acid with 5% acetonitrile before loading onto a 2 cm EASY-column

(Thermo Scientific) coupled to an in-house made nano HPLC column (20 cm × 75 um) packed with LUNA C18 media (Phenomenex). Analysis was performed on a Velos Pro mass spectrometer (Thermo Scientific) operated in data-dependent mode using 120-min gradients in EASY-LC system (Proxeon) with 95% water, 5% acetonitrile (ACN), 0.1% formic acid (FA) (solvent A), and 95% ACN, 5% water, 0.1% FA (solvent B) at a flow rate of 220 nl min$^{-1}$. The acquisition cycle consisted of a survey MS scan in the normal mode followed by 12 data-dependent MS/MS scans acquired in the rapid mode. Charge state was not recorded. Dynamic exclusion was used with the following parameters: exclusion size 500, repeat count 1, repeat duration 10 s, exclusion time 45 s. Target value was set at $10^4$ for tandem MS scan. The precursor isolation window was set at 2 $m/z$. The complete analysis comprised two independent biological replicates.

**Mass spectrometry data analysis.** The resulting spectrum files were transformed into MGF format by MSConvert software and interrogated by MASCOT 2.4 search engine using human UniProt database version 15 concatenated with reverse sequences for estimation of false discovery rate (FDR) and with a list of common contaminants (40,729 entries in total). The search parameters were as follows: full tryptic search, 2 allowed missed cleavages, peptide charges +2 and +3 only, MS tolerance 1 Da, MS/MS tolerance 0.5 Da. Permanent post-translational modification was: cysteine carbamidomethylation. Variable post-translational modifications were: protein N-terminal acetylation, Met oxidation and N-terminal Glutamine to pyro-Glutamate conversion. The remaining analysis was performed as previously described[54]. To summarize, the minimal ion score threshold was chosen such that a peptide FDR below 1% was achieved. The peptide FDR was calculated as: 2 × (decoy_hits)/(target + decoy hits). The mass spectrometry proteomics data have been deposited to the ProteomeXchange Consortium[55] via the PRIDE partner repository with the data set identifier PXD002421 and 10.6019/PXD002421. Spectral counts for all detected proteins were assembled using an in-house written Python script. The adjustment of spectral counts was done as previously described[54].

**RNA immunoprecipitation.** RIP assays were performed using a Millipore EZ-Magna RIP RNA-Binding Protein Immunoprecipitation kit (Millipore, #17-700) according to the manufacturer's instructions. RIP-PCR was performed as qPCR, as described above, using total RNA as input controls. 1:150th of RIP RNA product was used per PCR reaction. Antibodies used for RIP are listed in Supplementary Table 4. All RIP assays were performed in biological duplicate.

**Invasion assay.** $3 \times 10^5$ cells were seeded in a 24-well corning FluoroBlok chamber pre-coated with Matrigel (BD Biosciences). Medium containing 10% FBS in the lower chamber served as chemoattractant. After 48 h, cells remaining on the lower side of the membrane were stained with calcein AM (C34852 invitrogen). The invasive cells adhering to the bottom surface of the filter were quantified under a fluorescent microscope ( × 2).

**Antibodies and immunoblot analyses.** Western immunoblot assays were performed by running cell lysates on 4–12% SDS polyacrylamide gels (Novex) to separate proteins. Proteins were then transferred to a nitrocellulose membrane (Novex) via wet transfer at 30 V overnight. Blocking buffer incubation was then performed for 1 h (Tris-buffered saline, 0.1% Tween (TBS-T), 5% nonfat dry milk). Indicated antibodies were then added to membrane and incubated at 4 °C overnight. Enhanced chemiluminescence (ECL Prime) was utilized to develop blots via the manufacturer's protocol. All the antibodies used in this study are described in Supplementary Table 4. Representative full blot images are shown in Supplementary Fig. 8.

**Chromatin immunoprecipitation.** HighCell# ChIP kit (Diagenode) was utilized to perform ChIP assays via the manufacturer's protocol. Briefly, MCF7 cells were grown in charcoal-stripped serum media (described above) for 72 h and then stimulated 10 nM estradiol for 12 h. Cells were then crosslinked using 1% formaldehyde for 10 min, and crosslinking was quenched for 5 min at room temperature using a 1/10 volume of 1.25 M glycine. Cells were then lysed and sonicated (Bioruptor, Diagenode), yielding an average chromatin fragment size of 300 bp. An equivalent amount of chromatin equivalent to $5 \times 10^6$ cells was used for the ChIP for all antibodies. DNA bound to immunoprecipitated product was isolated (IPure Kit, Diagenode) via overnight incubation with antibody at 4 °C. Samples were then washed, and crosslinked reversed.

**ChIP-seq library construction and sequencing analysis.** DNA was purified for library preparation using the IPure Kit (Diagenode). The ChIP-seq sample preparation for sequencing was performed according to the manufacturer's instructions (Illumina). ChIP-enriched DNA samples (1–10 ng) were converted into blunt-ended fragments using T4 DNA polymerase, Escherichia coli DNA polymerase I large fragment (Klenow polymerase) and T4 polynucleotide kinase (New England BioLabs (NEB)). A single adenine base was added to fragment ends by Klenow fragment (3′ to 5′ exo⁻; NEB) followed by ligation of Illumina adaptors

(Quick ligase, NEB). The adaptor-modified DNA fragments were enriched by PCR using the Illumina Barcode primers and Phusion DNA polymerase (NEB). PCR products were size selected using 3% NuSieve agarose gels (Lonza) followed by gel extraction using QIAEX II reagents (QIAGEN). Libraries were quantified with the Bioanalyzer 2100 (Agilent) and sequenced on the Illumina HiSeq 2000 Sequencer (100-nucleotide read length). ChIP-seq data were mapped to human genome version hg19 using BWA[56]. The MACS program[57] was used to generate coverage map files to visualize the raw signal on the UCSC genome browser[58]. Hpeak[59], a hidden Markov model (HMM)-based peak-calling software program designed for the identification of protein-interactive genomic regions, was used for ChIP-seq peak determination.

**ChIP-seq peak promoter overlap.** Overlap of ChIP-seq peaks with gene promoters was performed using the BEDTools 'coverage' tool. Intervals of ± 5–10 kilobases surrounding unique transcriptional starts were used to assess promoter overlap.

**Coding potential scoring.** Coding potential for all lncRNA transcripts was determined as described previously[4]. The alignment-free Coding Potential Assessment Tool (CPAT)[25] was used to determine coding probability for each transcript. CPAT determines the coding probability of transcript sequences using a logistic regression model built from ORF size, Fickett TESTCODE statistic, and hexamer usage bias.

**Xenograft analysis.** All experimental procedures were approved by the University of Michigan Committee for the Use and Care of Animals (UCUCA) and conform to all regulatory standards. A total of $5 \times 10^6$ cells of T47D control or T47D shM41 cells were suspended in 100 ul of PBS/Matrigel (1:1) were injected subcutaneously in 5-week-old pathogen-free female CB-17 severe combine immunodeficient mice (CB-17 SCID) which simultaneously received a 60-day slow release pellet containing 0.18 mg of 17b-estradiol (Innovative Research of America). Tumours were measured weekly using a digital caliper. Growth in tumour volume was recorded using digital calipers and tumour volumes were estimated using the formula ($\pi/6$) $(L \times W^2)$, where $L$ = length of tumour and $W$ = width. In addition, mouse livers were collected to determine spontaneous metastasis by measuring human Alu sequence. Briefly, genomic DNA from livers were prepared using Puregene DNA purification system (Qiagen), followed by quantification of human Alu sequence by human Alu specific Fluorogenic Taqman qPCR probes.

**RNA-seq data processing.** Sequence quality control was done using FASTQC (http://www.bioinformatics.babraham.ac.uk/projects/fastqc). Next, reads mapping to mitochondrial DNA, ribosomal RNA, poly-A, poly-C, Illumina sequencing adaptors, and the spiked-in phiX174 viral genome were filtered. Sequences were downloaded from the Illumina iGenomes server (2012, March 9). Mapping was performed using bowtie2 (2.0.2). Reads were mapped using TopHat2 (2.0.6 and 2.0.8) using default parameters. A human genome reference was constructed from UCSC version Feb 2009 (GRCh37/hg19) chromosomes 1–22, X, Y and mitochondrial DNA, and references from alternate haplotype alleles were omitted. Bowtie-build and bowtie2-build were used to build genome reference for Bowtie versions 0.12.8 and 2.0.2 were, respectively. The Ensembl version 69 transcriptome was used as a reference gene set. Using the --transcriptome-index option in TopHat version 2.0.6 (ref. 60), alignment index files were prepared from this reference for Bowtie versions 0.12.8 and 2.0.2.

**RNA-seq transcript expression estimation.** Cufflinks version 2.1.1 (ref. 61) was used with the following parameters to estimate transcript abundance from RNA-seq data: '--max-frag-multihits = 1', '--no-effective-length-correction', '--max-bundle-length 5000000', '--max-bundle-frags 20000000'. To convert FPKM abundance estimates (generated by Cufflinks) to approximate fragment count values we multiplied each FPKM by the transcript length (in kilobases) and by the 'Map Mass' value (divided by 1.0E⁶) found in the Cufflinks log files.

**Breast cancer tissue expression heatmap generation.** The 'gplots' R-package was used to generate heatmaps using the heatmap.2 function. For the cancer versus normal heatmap, expression was normalized as log2 of the fold change over the median of the normal samples for each transcript. For the ER-positive versus ER-negative heatmap, expression was normalized to the median of the ER-negative samples. Unsupervised heirarchical clustering was performed with the hclust function, using Pearson correlation as the clustering distance, using the 'ward' agglomeration method.

**RNA-seq differential expression testing.** Differential expression testing was performed using the SSEA tool described previously[7]. Briefly, following count data normalization, SSEA performs the weighted KS-test procedure described in GSEA[42]. The resulting enrichment score statistic describes the enrichment of the sample set among all samples being tested. To test for significance, SSEA enrichment tests are performed following random shuffling of the sample labels.

These shuffled enrichment tests are used to derive a set of null enrichment scores (1,000 null enrichment scores computed). The nominal $P$ value reported is the relative rank of the observed enrichment score within the null enrichment scores. Multiple hypothesis testing is performed by comparing the enrichment score of the test to the null normalized enrichment score distributions for all transcripts in a sample set. This null normalized enrichment score distribution is used to compute FDR $Q$ values in the same manner used by GSEA[42].

**Associations with oncomine clinical signatures.** We identified the top 150 positively and negatively correlated genes (Spearman's correlation) to *DSCAM-AS1* among the ER-positive breast cancer samples. These gene lists were imported into Oncomine[27] as custom concepts. We then nominated significantly associated breast cancer concepts with odds ratio $> 4.0$ for negatively associated concepts and $> 6.0$ for positively associated concepts and $P$ value $< 1 \times 10^{-6}$. Nodes and edges of these associations were exported and a concept association network was generated using Cytoscape version 3.2.1. Node positions were computed using the Force-Directed Layout algorithm in Cytoscape using the odds ratio as the edge weight. Node positions were subtly altered manually to enable better visualization of node labels.

**Association of correlation signatures with oncomine concepts.** Correlation analysis described above was performed for *DSCAM-AS1, EZH2, HOTAIR, MALAT1,* and *NEAT1*. For each gene, we created a signature of the top 150 most positively and top 150 most negatively correlated genes. We performed a Fisher's exact test of overlap for each of the above gene signatures with Oncomine clinical signatures for cancer versus normal, clinical recurrence, clinical survival, metastasis, and high clinical grade. The following studies were utilized: Curtis Breast[26], Ma Breas[62], TCGA Breast[28], Zhao Breast[29], Bittner Breast[63], Desmedt Breast[30], Ivshina Breast[31], Loi Breast[32], Lu Breast[33], Perou Breast[22], Schmidt Breast[34], Sorlie Breast[23], vantVeer Breast[64], Wang Breast[36], Boersma Breast[37], Kao Breast[38], Symmans Breast[39] and vandeVijver Breast[40]. For each Oncomine concept, overlap was tested for the top 1, 5 and 10% of genes up- and downregulated, and the gene signature with the greatest odds ratio was selected for each study. Signature comparisons were performed using a one-sided Fisher's exact test.

**Survival analysis with TCGA breast data.** Association of DSCAM-AS1 levels on clinical outcomes was assessed using the TCGA breast cohort. Survival data was obtained from the TCGA data portal. ER-positive samples were used for survival analysis as indicated by the TCGA clinical metadata via IHC status. Samples with DSCAM-AS1 expression $> 10$ FPKM were grouped into the 'DSCAM-AS1 high' category and samples with expression $< 1$ FPKM were grouped into the 'DSCAM-AS1 low' category. Kaplan-Meier analysis was performed, and log-rank test was performed to assess statistical significance.

**Tissue expression level percentile metric.** To generate a metric to summarize the expression of each lncRNA in breast cancer tissues, we identified the expression level of the 95th percentile sample among all breast RNA-seq samples including cancers tissue, normal tissue, and cell lines.

**RNA-sequencing library preparation.** Total RNA was obtained from cancer cell lines, and RNA quality was determined using an Agilent Bioanalyzer. Poly-A transcriptome libraries from the mRNA fractions were generated following the Illumina RNA-seq protocol. Each sample was sequenced in a single lane with the Illumina HiSeq 2000 (100-nucleotide read length) as previously described[3,65]. The dUTP method of second-strand marking was used for strand-specific library preparation as described previously[66].

**Gene set enrichment analysis.** Expression levels of *DSCAM-AS1* were correlated (Spearman) to the expression of all protein-coding genes across all ER-positive breast cancers. The protein-coding genes were then ranked by the Spearman Rho value, and used in a weighted, preranked GSEA analysis against MSigDB gene sets V5.0 (ref. 67).

***In silico* binding prediction.** To obtain potential HNRNPL binding sites on DSCAM-AS1, we utilized GraphProt[68] to learn a predictive model from genome-wide HNRNPL binding sites identified by iCLIP-seq[48]. For training data generation, we extracted the genomic binding positions (GSE37560) with BED table scores $> = 10$, followed by an extension of $\pm 20$ nt resulting in 41 nt long binding sites. After mapping the sites to annotated RefSeq genes obtained from UCSC, an equally-sized set of negative sites was selected such that the sites were on the same RefSeq genes and did not overlap with any of the identified positive sites from the initial table. The GraphProt sequence model trained on these data was then used to identify high-scoring sites in the *DSCAM-AS1* sequence (NCBI GenBank NR_038899.1). The highest-scoring site centred at RNA position 923 contains a CA-repeat motif known for its affinity towards HNRNPL and was thus used for subsequent analysis.

**Rapid amplification of cDNA ends (RACE).** 5′ and 3′ RACE was performed using the GeneRacer RLM-RACE kit (Invitrogen) according to the manufacturer's protocols. RACE PCR products obtained using Platinum Taq high-fidelity polymerase (Invitrogen), were resolved on a 1.5% agarose gel. Individual bands were gel purified using a Gel Extraction kit (Qiagen), and cloned into PCR4 TOPO vector, and sequenced using M13 primers.

**Single-molecule fluorescence *in situ* hybridization.** Single-molecule fluorescence *in situ* hybridization was performed as described[69], with some minor modifications. Cells were grown on 8-well chambered coverglasses, formaldehyde fixed and permeablized overnight at 4 °C using 70% ethanol. Cells were rehydrated in a solution containing 10% formamide and $2 \times$ SSC for 5 min and then treated with 10 nM fluorescence *in situ* hybridization probes for 16 h in $2 \times$ SSC containing 10% dextran sulfate, 2 mM vanadyl-ribonucleoside complex, 0.02% RNAse-free BSA, 1 µg µl$^{-1}$ *E. coli* transfer RNA and 10% formamide at 37 °C. After hybridization, cells were washed twice for 30 min at 37 °C using a wash buffer (10% formamide in $2 \times$ SSC). Cells were then mounted in solution containing 10 mM Tris/HCl pH 7.5, $2 \times$ SSC, 2 mM trolox, 50 µM protocatechiuc acid and 50 nM protocatechuate dehydrogenase. fluorescence *in situ* hybridization samples were imaged in three dimensions using HILO illumination as described[70]. Images were processed using custom-written macros in ImageJ. Analysis routines comprises 3 major steps: background subtraction, Laplacian of Gaussian (LoG) filtering and thresholding. Spots with intensity above set threshold are represented in images. Probes were designed to target all isoforms of the DSCAM-AS1 transcript. Probe sequences targeting DSCAM-AS1 (21 probes per transcript) are as follows: 5′-cctatcccttttctctaagaa-3′, 5′-acttctgcaaaaacgtgctg-3′, 5′-ggttccactccatt ttaatt-3′, 5′-ctatagcgtcttatcagctg-3′, 5′-catgtgtccggatatcattt-3′, 5′-tcagtgagtggataact ggt-3′, 5′-aattctagtggaggcaccta-3′, 5′-ctaagtagcttcatctttcc-3′, 5′-caactgcgtgtttccta gtc-3′, 5′-agcattctctgttttaacca-3′, 5′-ttagcaactgccttgctctg-3′, 5′-gctgtccagtttagta aca-3′, 5′-cgttgtgagcctgagagatc-3′, 5′-agaacttccctagaggagtg-3′, 5′-atggggagtgagaccaa aca-3′, 5′-tggaggagggacagagaagg, 5′-tgtgggtgattggtactttt-3′, 5′-atggatgagtatgtcat gcc-3′, 5′-tattgccatggttagcatga-3′, 5′-aatgcatgcttgatggagct-3′.

**Data availability.** Sequence data that support the findings of this study have been deposited in the Short Read Archive with the accession code SRP078392. Tissue ChIP-seq data referenced in this study are available in the Gene Expression Omnibus with the accession code GSE32222. All remaining data are contained within the Article and Supplementary Information files or available from the author on request.

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

## Acknowledgements

We thank K.W.-R. for help with the mouse Xenograft work, the University of Michigan DNA Sequencing Core for Sanger sequencing and K. Giles for critically reading the manuscript and for the submission of documents. This work was supported in part by US National Institutes of Health Prostate Specialized Program of Research Excellence grant P50CA186786, Early Detection Research Network grant UO1 CA111275, US National Institutes of Health grants R01 CA132874 and RO1 CA154365 (A.M.C.), and US Department of Defense grant PC100171 (A.M.C.). A.M.C. is supported by the Prostate Cancer Foundation and the Howard Hughes Medical Institute. A.M.C. is an American Cancer Society Research Professor and a Taubman Scholar of the University of Michigan. R.M. was supported by a Prostate Cancer Foundation Young Investigator Award and by US Department of Defense Post-Doctoral Fellowship W81XWH-13-1-0284. Y.S.N. is supported by a University of Michigan Cellular and Molecular Biology National Research Service Award Institutional Predoctoral Training Grant.

## Author contributions

Y.S.N., S.H., A.M.C. and F.Y.F. conceived the study and analyses. Y.S.N. performed all bioinformatics analyses and guided study design with the assistance of M.K.I. and S.G.Z. S.H. performed cellular and molecular biology experiments with the assistance T.M., R.M., C.Z., J.R.P., Y.H., S.R., S.K., R.F.S., L.X. and U.S., K.W.-R. performed mouse xenograft work. X.C. generated RNA-seq libraries and performed the sequencing. A.P. performed the mass spectrometry. M.U., A.G., C.C., R.B. and C.S.S. performed the *in silico* binding predictions. J.M.R. produced the tamoxifen resistance cell line. D.F.H. and L.J.P. provided guidance and analysis for the clinical work. Y.S.N., A.M.C.

and F.Y.F. wrote the manuscript. All authors discussed results and commented on the manuscript.

## Additional information

**Competing financial interests:** Oncomine is supported by ThermoFisher, Inc. (Previously Life Technologies and Compendia Biosciences). A.M.C. was a co-founder of Compendia Biosciences and served on the scientific advisory board of Life Technologies before it was acquired. D.F.H. receives research funding from Astra-Zeneca, Eli Lilly, Janssen, Pfizer and Puma. He has stock options in OncImmune and in InBiomotion, and he is named as a primary inventor on two patents issued to the University of Michigan, one of which is licensed to Janssen and for which he receives royalties. These patents are not pertinent to the work presented here. The remaining authors declare no competing financial interests.

