## [Peer Review File · Nature Communications]

Reviewer #1 (Remarks to the Author):

The authors have fully addressed most of my previous concerns. In consequence of the additional analyses, one additional change needs to be included into the manuscript: Since the authors did not find any direct association of DSCAM-AS1 with the clinical parameters in the TCGA dataset, the respective finding needs to be included and discussed in the manuscript in addition to the analysis presented in lines 151 - 177.

Reviewer #2 (Remarks to the Author):

The authors have addressed my previous concerns and I am satisfied that this paper makes an important contribution and should be published.

Reviewer #3 (Remarks to the Author):

NCOMMS-16-12203-T by Niknafs et al. (Chinnaiyan)

"The lncRNA landscape of breast cancer reveals the role of BRCAT431 in breast cancer progression and tamoxifen resistance"

Summary

In the revised version of their paper exploring the role of lncRNAs in breast cancers, the authors have improved the manuscript by clarifying the details of their analyses, citing relevant references, dealing with issues of nomenclature, highlighting their broad analysis across many breast cancer samples, adding some control experiments, completing a more careful annotation of the DSCAM-AS1 gene and transcript, including statistical analyses, etc.

Overall Comments

The authors have addressed most of my original concerns. The revised manuscript has been improved considerably. Although the role of DSCAM-AS1 in ER+ cancers is somewhat clearer now, future investigations are required to explore it as a potential therapeutic target.

Response to reviewers:

Reviewer #1 (Remarks to the Author):

The authors have fully addressed most of my previous concerns. In consequence of the additional analyses, one additional change needs to be included into the manuscript: Since the authors did not find any direct association of DSCAM-AS1 with the clinical parameters in the TCGA dataset, the respective finding needs to be included and discussed in the manuscript in addition to the analysis presented in lines 151 - 177.

>> We thank reviewer one for accepting our changes, and agree that the TCGA dataset is a necessary addition. We have altered the manuscript accordingly, and these data now appear in Supplementary Fig. 3d. We have also added manuscript text to discuss this, and have added appropriate methods.

Reviewer #2 (Remarks to the Author):

The authors have addressed my previous concerns and I am satisfied that this paper makes an important contribution and should be published.

>> We appreciate the reviewer's effort put into our manuscript, and appreciate the comments that have improved our study throughout the review process.

Reviewer #3 (Remarks to the Author):

NCOMMS-16-12203-T by Niknafs et al. (Chinnaiyan)

"The lncRNA landscape of breast cancer reveals the role of BRCAT431 in breast cancer progression and tamoxifen resistance"

Summary

In the revised version of their paper exploring the role of lncRNAs in breast cancers, the authors have improved the manuscript by clarifying the details of their analyses, citing relevant references, dealing with issues of nomenclature, highlighting their broad analysis across many breast cancer samples, adding some control experiments, completing a more careful annotation of the DSCAM-AS1 gene and transcript, including statistical analyses, etc.

>> The reviewer's overall positive assessment of our rebuttal is appreciated. All of the comments and suggestions were helpful in improving our manuscript significantly.

Overall Comments

The authors have addressed most of my original concerns. The revised manuscript has been improved considerably. Although the role of DSCAM-AS1 in ER+ cancers is somewhat clearer now, future investigations are required to explore it as a potential therapeutic target.

>> We agree that this study is a first step towards further understanding this gene and its role in breast cancer.